# Psychological and Physical Health of Organic and Conventional Farmers: A Review

Lucas David [1], Michaël Dambrun [1], Rosie Harrington [1], Michel Streith [1] and Audrey Michaud [2,*]

1   Centre National de la Recherche Scientifique, Laboratoire de Psychologie Sociale et Cognitive, Université Clermont Auvergne, 63000 Clermont-Ferrand, France; lucas.david@etu.uca.fr (L.D.); michael.dambrun@uca.fr (M.D.); rose.harrington@uca.fr (R.H.); michel.streith@uca.fr (M.S.)
2   UCA, INRAE UMR Herbivores, 63 122 Saint Genès Champanelle, France
*   Correspondence: audrey.michaud@vetagro-sup.fr

**Abstract:** Farmers' health compared to the general population has been the object of some studies and reviews. Among all factors implied in psychological and physical health, the farming system (i.e., organic or conventional farming) was identified as one of the relevant factors to investigate. This article aims to review the literature established on the comparison between organic and conventional farmers' health and its correlates. Twenty-nine quantitative or qualitative articles were identified for inclusion (*n* = 29). Results showed that organic farmers had globally better psychological and physical health than conventional farmers. Effect sizes were small to large; they differed according to the target outcome. In addition, factors correlated with farmers' health were usually psychological, social, financial, and agricultural. This review of literature encourages further research in this area, particularly on developing agricultural models.

**Keywords:** organic farming; conventional farming; health

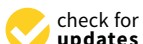


## 1. Introduction

The agricultural sector, which is currently undergoing numerous crises (environmental, economic, etc.) [1–3], is also sensitive in terms of the farmers' health. In fact, numerous studies and reviews highlight health problems in this sector, particularly physical and psychological problems [4,5]. Health problems can lead to a higher suicide rate [6], a lower score on the General Health Questionnaire (GHQ-12) compared to non-farmers [7], and a higher cause of death by cancer or other illnesses [5].

Numerous predictors and risk factors had been listed to explain these indicators, such as the high prevalence of stress, depression, suicide, or physical health problems among farmers, and concerning pesticide exposure, financial difficulties, climate variabilities, and poor physical health [4]. Depression and suicide are associated with factors such as gender, marital status, financial decline, perceived health and general health status, previous injury, stress, safety knowledge and behavior, pesticides, farming hazards, year of farming, indebtedness, lack of financial investment, poor irrigation, use of chemical fertilizer, or even stress, family responsibilities, work satisfaction, and work effects on private life [8–11]. Focusing on stress, Truchot and Andela (2018) listed eight variables that correlated with the Maslach Burnout Inventory (MBI) and the Beck Hopelessness Scale (BHC), including workload, conflicts with colleagues, or family members, and financial worries [12]. More specifically, self-reported distress and generalized anxiety scores were positively correlated with climate risk perception and climate harm [13]. These health problems concern agriculture and particularly the conventional model.

Organic farming is often mentioned as a solution to these health problems, such as pesticides [14]. Organic agriculture favors prevention and management practices that respect natural balances rather than the use of interventions and production factors of external origin derived from synthetic chemistry. The reasons for converting to this agricultural

model vary according to the farmer (economic reasons and environmentally-conscious), and the behavior of farmers towards this agricultural model is also variable [15,16]. Of the many factors that may affect farmers' mental and physical health (e.g., working condition, localization . . . ), the type of model—organic/conventional farming—emerges. The organic agricultural model effects on physical and/or psychological health were shown [17,18]. Mattila et al. (2020) indicated a lower work ability score for organic farmers [19]. The risk aversion was also lower for organic farmers [20,21]. McCann et al. (1997) showed that organic farmers could take more risks for a long-term benefit [22]. In the same paper, pollution was perceived as a bigger threat to human health by organic farmers than conventional farmers. The effects of agricultural models on the psychological and physical farmers' health remain, however, unclear and require a synthesis of existing literature.

The aim of this paper is to sum up the literature established on this topic. The main objective is to make an inventory of articles that compare organic, agroecological, and conventional farmers' physical and/or psychological health. Additionally, results concerning correlates of both kinds of farmers' health will be collected. This synthesis will highlight a possible effect of the agricultural model on farmers' health.

## 2. Method

### 2.1. Selection Method

In this review, included articles had to focus on the comparison between organic and conventional farmers' psychological or physical health. Agroecological systems are integrated into organic systems. AS these systems differ in their functioning from organic agriculture (optimizing the management of cycles on the farm, minimizing the use of even natural inputs...), in the literature, few articles on agroecology have been identified, and those studied are conducted in a very similar way to organic agriculture. They have therefore been grouped together. No exclusion criteria were established for publication date, location, or publication language.

Searches were conducted in Pubmed, PsychInfo, and Google Scholar. The following key words were used: "Conventional vs. organic"; "Conventional vs. organic + health"; "Conventional vs. organic + well-being"; "Conventional vs. organic + burnout"; "Conventional vs. organic + suicide"; "Organic vs. conventional"; "Organic vs. conventional + health"; "Organic vs. conventional + well-being"; "Organic vs. conventional + burnout"; "Organic vs. conventional + suicide"; "Agroecology vs. conventional"; "Agroecology vs. conventional + health"; "Agroecology vs. conventional + well-being"; "Agroecology vs. conventional + burnout"; "Agroecology vs. conventional + suicide"; "Organic farming"; "Organic farming + health"; "Organic farming + farmer health"; "Organic farming + well-being"; "Organic farming + burnout"; "Organic farming + suicide"; "Conventional farming"; "Conventional farming + health"; "Conventional farming + farmer health"; "Conventional farming + well-being"; "Conventional farming + burnout"; "Conventional farming + suicide"; "Agroecology"; "Agroecology + health"; "Agroecology + farmer health"; "Agroecology + well-being"; "Agroecology + burnout"; "Agroecology + suicide".

Figure 1 summarizes the process of identification and selection of items according to the PRISMA procedure. Articles excluded from these results did not compare organic and conventional farmers (e.g., comparison between farmers and non-farmers; *n* = 17) or did not focus on psychological or physical health (*n* = 9). Articles that focused on only organic or conventional health were kept. The final sample contained 29 papers, including 18 quantitative articles, 8 qualitative articles, 1 quantitative treated as qualitative because of lack of statistical data, and 2 papers that contained both quantitative and qualitative data. In this sample, 21 articles only focused on the comparison between organic and conventional farmers' health, 5 treated on correlates of only organic or conventional farmers' health without comparison, and 3 gave data for these two themes.

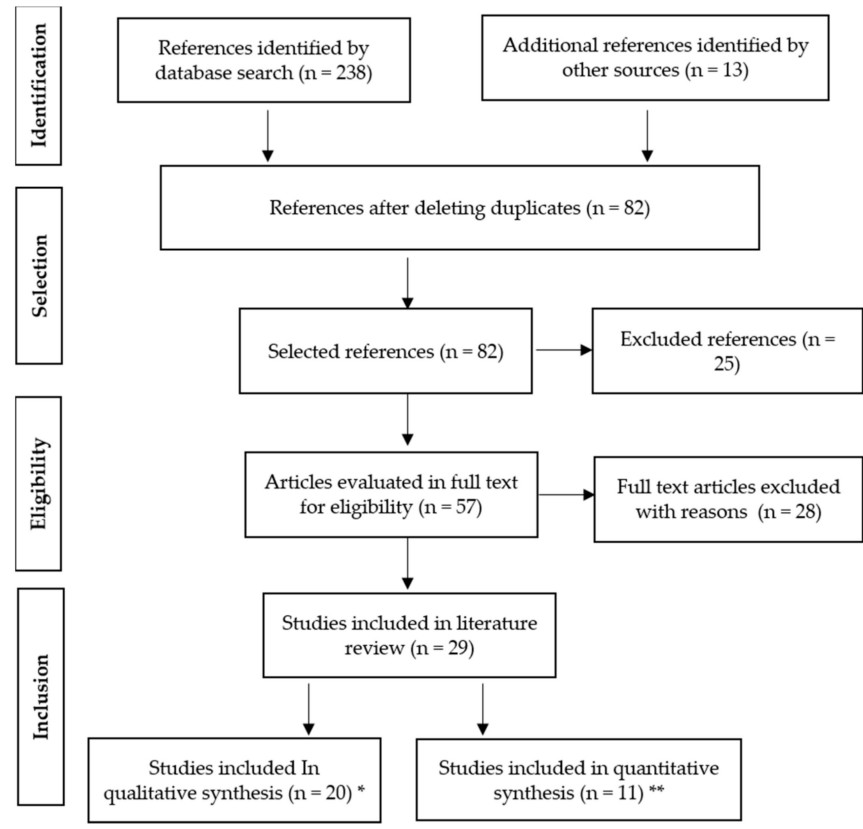

**Figure 1.** Procedure of articles selection according to PRISMA guidelines. *Notes*. * As explained, 18 articles were quantitative and 2 were both quantitative and qualitative, hence n = 20. ** In addition, 8 articles were qualitative, 1 was quantitative but treated as qualitative, and 2 were both quantitative and qualitative, hence n = 11.

*2.2. Statistical Procedure*

Statistical analyses consisted of calculating Cohen's *d*, a statistical parameter that gives an estimation of effect sizes [23,24]. In this review, Cohen's *d* was useful to have an idea of the importance of each effect. Several procedures were necessary to calculate effect sizes due to the diversity of statistics used in studies (a single method could not be applied to all studies). Effect sizes were grouped into three categories defined by Cohen (1988): small for $0.20 \leq d < 0.50$; medium for $0.50 \leq d < 0.80$; and large for $d \geq 0.80$ [24].

Sixty-three Cohen's *d* were obtained for qualitative results using online calculators [23,25]. Mean Cohen's *d* per study or per category were not calculated because of the lack of statistical parameters in some papers and the heterogeneity of the measures. Eight procedures were carried out using absolute values of parameters presented in articles.

In Procedure 1, standardized β coefficients were converted into r with Peterson and Brown's (2005) method [26], and r was converted into Cohen's *d* (calculators 12 and 14 [23]). This procedure gave us 12 effect sizes from four studies. In Procedure 2, Fisher's F and sample sizes were used to obtain Cohen's *d* (calculator 6; [23]). This procedure gave us 3 effect sizes from two studies. In Procedure 3, odds ratios were converted into Cohen's *d*'s (calculator 14; [23]). This procedure gave us 12 effect sizes from three studies. In Procedure 4, the risks ratio was transformed into an odds ratio [25] and then the odds ratios into Cohen's *d* (calculator 14; [23]). This procedure gave us 23 effect sizes from two studies. In Procedure 5, the t-statistic of a dependent test, sample size, and r were used to obtain Cohen's *d* (calculator 5; [23]). This procedure gave us 2 effect sizes from one study. In Procedure 6, chi-square parameter and sample size were used to obtain Cohen's *d* (calculator 15; [23]). This procedure gave us 1 effect size from one study. In Procedure 7,

r was converted to Cohen's *d* (calculator 14, [23]). This procedure gave us 7 effect sizes from two studies. In Procedure 8, the standard deviation of each group was calculated using sample size and confidence interval [27]. Then Cohen's *d* was obtained with standard deviation, mean, and sample size (calculator 2; [23]). This procedure gave us 3 effect sizes from one study. For six studies, it was not possible to calculate Cohen's *d* because the statistics presented were not sufficient. For these studies, we only reported the direction of effects and *p*-values.

## 3. Results

### 3.1. Health of Organic and Conventional Farmers: A Review of Quantitative Data

3.1.1. Psychological Health

Seven studies presented quantitative psychological health data. Depression scores were obtained with different scales or items. The Short Depression Happiness Scale (SDHS) was used by Cross et al. (2008), the Center for Epidemiologic Studies Depression Scale (CESD) was used by Khan et al. (2018) [17,28]. Kaufman (2015) and Nankongnab et al. (2020) measured depression with specific items [29,30]. Some dimensions of mental health were measured by Cross et al. (2008) with the SF-36, the EQ-D5, and the Visual Analogue Scale (VAS) [28]. Two studies focused on satisfaction; life satisfaction for Mzoughi (2014) and satisfaction with farm work for Rickson et al. (1999) [31,32]. In addition, Khan et al. (2018) evaluated neurological symptoms, including behavioral and cognitive symptoms, Kaufman (2015) evaluated sadness and anger, and Howard et al. (2020) had a measure of climate-related anxiety [13,17,29].

A total of 2266 farmers participated in these studies. Nine hundred and forty-seven of them were organic farmers, and 1319 were conventional farmers. Participants were located in different countries (e.g., UK, USA, Thailand).

Some results showed differences between organic and conventional farmers' psychological health. As depicted in Table 1, organic farmers obtained lower depression scores on the SDHS than conventional farmers [28]. Kaufman (2015) found organic farmers less depressed, sad, and angry [29]. Fewer neurological symptoms and behavioral symptoms were identified among organic farmers [17]. They also had less climate-related anxiety than their conventional colleagues [13]. Organic farmers were more life-satisfied and satisfied with farm work than conventional farmers [31,32]. However, Nankongnab et al. (2020) did not find a significant difference between organic and conventional farmers for depression [30], and these groups did not differ either in their CESD score or in their cognitive symptoms [17]. In addition, the type of farming did not affect the SF-36, EQ-D5, and VAS scores [28].

To sum up, some studies found differences between organic and conventional farmers' psychological health, especially concerning depression and satisfaction. Identified effects always were at the advantage of organic farmers. Eight Cohen's *d* calculated in five studies varied from 0.25 to 0.76; thus, five effects identified were small, and three were moderate [24].

**Table 1.** Quantitative studies about organic vs. conventional farmers' psychological health.

| Authors | Location | Farming Production | Sample | Key Findings | Cohen's $d$ |
|---|---|---|---|---|---|
| Cross et al. (2008) [28] | UK | Vegetable production, horticulture | N = 605 Conventional: $n = 512$ Organic: $n = 93$ | Organic farmers were less depressed than conventional farmers on the SDHS ($p < 0.012$). No effect was found on the SF-36, the EQ-D5, or the VAS ($p > 0.10$). The farming method (i.e., organic vs. conventional farming) was a predictor of the Mental Component Score (MCS) ($p = 0.011$). | 0.37 (Farming method on the MCS) |
| Howard et al. (2020) [13] | USA-Montana | Beef cattle | N = 119 Conventional: $n = 77$ Organic: $n = 42$ | Organic farmers had less climate-related anxiety than conventional farmers ($p < 0.05$). | 0.45 (Farming method on anxiety) |
| Kaufman (2015) [29] | Thailand | Rice production | N = 139 Conventional: $n = 64$ Organic: $n = 75$ | Organic farmers were less sad, angry, or depressed than conventional farmers ($p < 0.05$). | 0.36 (Farming method on sadness, anger, and depression) |
| Khan et al. (2018) [17] | USA-Indiana | Not specified | N = 13706 Final sample: N = 357 Conventional: $n = 200$ Organic: $n = 157$ | Organic farmers had fewer total neurological symptoms ($p < 0.02$) and behavioral symptoms ($p < 0.03$) than conventional farmers. There was neither difference for cognitive symptoms nor for the total CESD depression score and for its constructs ($p > 0.10$). | 0.25 (Farming method on total neurological symptoms) 0.28 (Farming method on behavioral symptoms) |
| Mzoughi (2014) [31] | France-PACA | Wine growing, vegetable production, crop production | N = 280 Conventional: $n = 95$ Organic: $n = 185$ | Organic farmers (both recently and earlier) had more life satisfaction than conventional farmers ($p < 0.05$). | 0.75 (Organic farmers on satisfaction) 0.71 (Recently organic farmers on satisfaction) 0.76 (Earlier organic farmers on satisfaction) |
| Nankongnab et al. (2020) [30] | Thailand | Vegetable production | N = 478 Conventional: $n = 243$ Organic: $n = 235$ | No differences between organic and conventional farmers for depression ($p > 0.10$). | N/A (No significant) |
| Rickson et al. (1999) [32] | Australia | Crop and livestock production | N = 282 Conventional: $n = 122$ Organic: $n = 160$ | Organic farmers and their families were more satisfied with farm work ($p < 0.01$) and reported more control of farm work ($p < 0.05$) than conventional farmers. | N/A |

Cohen's $d$: 0.37 and 0.28 calculated with Procedure 1; 0.45 and 0.36 calculated with Procedure 2; 0.75, 0.71, and 0.76 calculated with Procedure 7; 0.25 obtained with Procedure 8.

3.1.2. Physical Health

Thirteen studies reported quantitative physical health data. Physiological parameters (e.g., thyroid hormones, cholesterol, lymphocytes percentages) were measured in five studies [33–37]. Physical health was evaluated with the SF-36, the EQ-D5, and the VAS by Cross et al. (2008) [28]. Hutter et al. (2021), Nankongnab et al. (2020), and Smit et al. (2006) focused on diverse physical symptoms (e.g., skin rash, wheezing) [18,30,38]. Neurological symptoms, including sensory and motor symptoms, were measured by Khan et al. (2018) [17]. Additionally, data about nutrition were collected by Deaconu et al. (2020) [39]. Kaufman (2015) used some items to evaluate health status, stamina, pain, and illness [29]. Finally, Setboonsarng and Lavado (2008) extracted the medical expenditures of farmers [40].

A total of 6074 farmers participated in these studies. There were 2392 organic farmers, 60 agroecological farmers, 37 ecological farmers, 3524 conventional farmers, and 61 participants were in a control group. Participants were from diverse countries (e.g., Portugal, UK, Thailand).

Several effects were found (see Table 2). Costa et al. (2014) noticed that both organic and conventional farmers had higher or lower physiological parameters than the control group [33]. Other indicators, such as body mass index (BMI), thyroids hormones, cholesterol, are globally lower among organic farmers compared to conventional farmers [34–37]. Within sixty-nine symptoms tested in five studies, ten were more frequent among organic farmers (e.g., chest pain, frequent urination), and nineteen were less present (e.g., sensory symptoms, wheezing with shortness of breath) [17,18,29,30,38]. In addition, agroecological farmers consumed more dairy and "other fruit" than reference farmers, without any difference for the other food categories [39]. Setboonsarng and Lavado (2008) showed lower medical expenditures for organic farmers than conventional farmers [40]. However, Cross et al. (2008) did not find differences between farmers on physical health scales [28].

AS general physical health measures did not differ between farmers, the results of these studies presented some differences for symptoms, pain, nutrition, or physiological indicators. These effects were in favor of organic farmers in ten studies [17,18,29,34–40]. Two studies presented effects in favor of each type of farmers, depending on symptoms [30,33]. Thirty-eight Cohen's *d* were calculated in six studies; they varied from 0.13 to 1.35. Two Cohen's *d* were inferior to 0.20, indicating no effect. Thirteen Cohen's *d* were between 0.20 and 0.50; then, these effects were small. Eleven were between 0.50 and 0.80, and twelve were equal or superior to 0.80, indicating these effects were respectively moderate and large [24].

**Table 2.** Quantitative studies about organic vs. conventional farmers' physical health.

| Authors | Location | Farming Production | Sample | Key Findings | Cohen's *d* |
|---|---|---|---|---|---|
| Costa et al. (2014, in press) [33] | Portugal | Horticulture | N = 182<br>Conventional: *n* = 85<br>Organic: *n* = 36<br>Control: *n* = 61 | Organic farmers had more CD56 and CD16 (natural killer cells) than conventional farmers and control group ($p < 0.01$). Conventional farmers had more OP-CRB (Urinary Organophosphates/Carbamates) ($p = 0.002$), THIO (Urinary Thioethers) ($p = 0.003$), MN-RET (Micronuclet in reticulocytes) ($p < 0.01$), total CA (number of aberrations) ($p < 0.01$), and CTA (Chromatoid type aberrations per 100 cells) ($p < 0.01$) than organic farmers and control group. Organic had less %T (percentage of tail DNA) ($p < 0.01$), conventional had more than control group ($p < 0.01$). Both had more MNL ((Micronuclet in binucleated cells) ($p < 0.05$) and less TCR-Mf (T-cell receptors mutation frequencies) ($p < 0.10$), and CD-19 (B lymphocytes) ($p < 0.05$) than control group. No difference for PYR (Urinary pyrethroids), BChE (Plasma butyrylcholtnesterase), and the percentage of CD3 (T lymphocytes), CD4 (T helper cells) and CD8 (T cytotoxic cells) ($p > 0.05$). | N/A |
| Cross et al. (2008) [28] | UK | Vegetable production, horticulture | N = 605<br>Conventional: *n* = 512<br>Organic: *n* = 93 | No effect found on the SF-36, the EQ-D5, or the VAS ($p > 0.10$). The farming method was not a predictor of the Physical Component Score (PCS). | N/A (No significant) |
| Deaconu et al. (2020, in press) [39] | Ecuador | Fruits, crop, and vegetable production | N = 90<br>Agroecology: *n* = 60<br>Conventional: *n* = 30 | Agroecological farmers consumed more dairy ($p = 0.0053$) and other fruits ($p = 0.0384$). There was no difference for grains, white roots and tuber, plantains, pulses, nuts and seeds, meat, eggs, green leafy vegetables, other vitamins A-rich fruit and vegetables, and other vegetables ($p > 0.10$). | 0.85<br>(Farming method on dairy consumption)<br>0.59<br>(Farming method on other fruits consumption) |
| Hutter et al. (2021) [38] | Ecuador | Banana production | N = 68<br>Ecology: *n* = 37<br>Conventional: *n* = 31 | Ecological farmers presented less dizziness ($p = 0.007$), nausea and vomiting ($p = 0.006$), strong fatigue ($p = 0.004$), diarrhea ($p = 0.043$), sleeplessness ($p = 0.025$), burning eyes ($p = 0.012$), skin irritations ($p = 0.035$), and irregular heartbeat ($p = 0.041$). There was no difference for headache, vision problems, excess salivation, exhaustion, stomach pain, runny nose, breathing difficulties, watering eyes, skin rashes, cough, twitches and trembling ($p > 0.05$). | 0.87<br>(Farming method on dizziness)<br>1.11<br>(Farming method on nausea and vomiting)<br>0.88<br>(Farming method on strong fatigue)<br>1.03<br>(Farming method on diarrhea)<br>0.67<br>(Farming method on sleeplessness)<br>0.78<br>(Farming method on burning eyes)<br>0.7<br>(Farming method on skin irritations)<br>0.96<br>(Farming method on irregular heartbeat) |
| Kaufman (2015) [29] | Thailand | Rice production | N = 139<br>Conventional: *n* = 64<br>Organic: *n* = 75 | Organic farmers had less pain or illness in the last three months than conventional farmers ($p < 0.05$). There was no difference in the health status in the last three years for stamina compared to other farmers or overall health status ($p > 0.10$). | 0.35<br>(Farming method on pain or illness in the last three months) |
| Khan et al. (2018) [17] | USA-Indiana | All (USDA online database) | N = 13706<br>Final sample: N = 357<br>Conventional: *n* = 200<br>Organic: *n* = 157 | Organic farmers had less total neurological symptoms ($p < 0.02$) and sensory symptoms ($p < 0.001$). There was no difference for motor symptoms ($p > 0.10$). | 0.21<br>(Farming method on total neurological symptoms)<br>0.39<br>(Farming method on sensory symptoms) |

**Table 2.** *Cont.*

| Authors | Location | Farming Production | Sample | Key Findings | Cohen's *d* |
|---|---|---|---|---|---|
| Kongtip et al. (2018) [34] | Thailand | Vegetable production | N = 436 Conventional: *n* = 216 Organic: *n* = 222 | Organic farmers had lower Body Mass Index (BMI) ($p = 0.005$), waist circumference ($p = 0.01$), and body fat percentage ($p = 0.018$) than conventional farmers. They had less triglyceride ($p = 0.045$), total cholesterol ($p < 0.001$), and LDL ($p < 0.001$), and more HDL ($p = 0.008$) than conventional farmers. No difference with blood glucose, blood pressure or metabolic syndrome ($p > 0.05$). | 0.44 (Farming method on BMI) 0.49 (Farming method on waist circumference) 0.8 (Farming method on body fat percentage) 0.6 (Farming method on triglyceride) 0.33 (Farming method on total cholesterol) 0.76 (Farming method on LDL) 0.87 (Farming method on HDL) |
| Kongtip et al. (2019) [35] | Thailand | Vegetable production | N = 417 Conventional: *n* = 195 Organic: *n* = 222 N = 438 Conventional: *n* = 213 | Organic farmers had a lower BMI ($p = 0.001$), lower levels of TSH ($p < 0.001$), FT3 ($p < 0.001$), T3 ($p < 0.001$), total cholesterol ($p < 0.001$), and LDL ($p < 0.001$), and more HDL ($p < 0.001$) than conventional farmers. No differences with FT4, T4, triglyceride, blood glucose, or blood pressure ($p > 0.10$). | N/A |
| Kongtip et al. (2020) [36] | Thailand | Vegetable production | Organic: *n* = 225 | Organic farmers had less total cholesterol ($p < 0.001$), HDL ($p < 0.001$), LDL ($p < 0.001$), blood glucose ($p = 0.009$), systolic blood pressure ($p = 0.002$), diastolic blood pressure ($p < 0.001$), and lower BMI ($p = 0.01$) and waist circumference ($p = 0.004$) than conventional farmers. No differences with triglyceride and body fat ($p > 0.10$). | N/A |
| Nankongnab et al. (2020) [37] | Thailand | Rice, vegetable, fruit, and sugarcane production | N = 438 Conventional: *n* = 213 Organic: *n* = 225 | Organic farmers had lower levels of TSH ($p < 0.001$), FT3 ($p < 0.001$), T4 ($p = 0.008$), T3 ($p < 0.001$), lower BMI ($p < 0.001$), less total cholesterol ($p < 0.001$), and LDL ($p < 0.001$), and more HDL ($p < 0.001$) than conventional farmers. No differences with FT4 or triglyceride ($p > 0.05$). | N/A |

**Table 2.** *Cont.*

| Authors | Location | Farming Production | Sample | Key Findings | Cohen's *d* |
|---|---|---|---|---|---|
| Nankongnab et al. (2020) [30] | Thailand | Vegetable production | N = 478 Conventional: *n* = 243 Organic: *n* = 235 | Organic farmers had less skin rash ($p = 0.023$), water blister ($p = 0.002$), loss of appetite/weight loss ($p = 0.001$), reduced hearing ability ($p = 0.001$), headache ($p < 0.001$), and dizziness ($p = 0.013$), and more urticarial skin ($p < 0.001$), chest pain ($p = 0.017$), mild fever in the afternoon/evening ($p = 0.006$), flatulence ($p < 0.001$), and frequent urination ($p = 0.027$) than conventional farmers. They also had more pain in the wrist/hand ($p < 0.001$), fingers ($p = 0.044$), upper back ($p < 0.001$), hip ($p = 0.001$), and ankle/feet ($p = 0.003$) than conventional farmers. No differences with skin itchy, skin dry and cracked, eye pain, blurred vision, irritated eyes, upper and lower limb weakness, jaundice, runny nose, cough, short breath, wheezing, neck, shoulder, elbow, and for lower back, and knee pain ($p > 0.10$). | 0.73 (Farming method on skin rash) 0.38 (Farming method on water blister) 0.3 (Farming method on loss of appetite/weight loss) 0.47 (Farming method on reduced hearing ability) 0.54 (Farming method on headache) 0.66 (Farming method on dizziness) 1.28 (Farming method on urticarial skin) 0.18 (Farming method on chest pain) 0.27 (Farming method on mild fever in the afternoon/evening) 0.32 (Farming method on flatulence) 0.13 (Farming method on frequent urination) 1.35 (Farming method on pain in the wrist/hand) 1.35 (Farming method on pain in fingers) 0.7 (Farming method on pain in upper back) 0.39 (Farming method on pain in hip) 0.87 (Farming method on pain in ankle/feet) |
| Setboonsarng and Lavado (2008) [40] | Thailand | Rice production | N = 626 Conventional: *n* = 317 Organic: *n* = 309 | Organic farmers had lower medical expenditures than conventional farmers ($p < 0.10$). | N/A |
| Smit et al. (2006) [18] | Netherlands | Livestock production, horticulture | N = 1798 Conventional: *n* = 1205 Organic: *n* = 593 | Organic farmers had less wheezing with shortness of breath (SOB) ($p < 0.05$) and woke less due to SOB ($p < 0.05$) than conventional farmers. No difference with daily cough, daily cough up phlegm, woken due to cough, wheezing, wheezing without a cold, asthma diagnosed by doctor, episode of asthma last year, use of drugs for asthma, asthma (ECRHS) ($p > 0.10$). | 0.2 (Farming method on wheezing with SOB) 0.51 (Farming method on woken due to SOB) |

Cohen's *d*: 0.85, 0.59, 0.87, 1.11, 0.88, 1.03, 0.67, 0.78, 0.70, 0.96, 0.20, and 0.51 calculated with Procedure 3; 0.35 calculated with Procedure 2; 0.44, 0.49, 0.80, 0.60, 0.33, 0.76, 0.87, 0.73, 0.38, 0.30, 0.47, 0.54, 0.66, 1.28, 0.18, 0.27, 0.32, 0.13, 1.35, 1.35, 0.70, 0.39, and 0.87 calculated with Procedure 4; 0.21 and 0.39 calculated with Procedure 8.

*3.2. Health of Organic and Conventional Farmers: A Review of Qualitative Data*

3.2.1. Psychological Health

Seven studies reported qualitative psychological health data. Altenbuchner et al. (2017) focused on females' perception of organic farming in comparison with conventional farming [41]. Bouttes et al. (2018; 2020) reported data about farmers' satisfaction, especially during conversion to organic farming [42,43]. Other satisfaction measures were collected by Sullivan et al. (1996), with well-being and stress measures [44]. In addition, values and well-being were evaluated in two studies [39,45]. Van Dam et al. (2010) made a study on the process of conversion to organic farming (i.e., attitude toward conventional practices and consequences of conversion) [46].

A total of 288 farmers participated in the studies. There were 57 organic farmers, 60 agroecological farmers, 48 conventional farmers, and 123 participants who were former conventional farmers who converted to organic. Participants came from different countries (e.g., India, France, USA).

As presented in Table 3, all studies reported differences between organic and conventional farmers. Bouttes et al. (2020) showed an increase in farmers' satisfaction during conversion to organic farming [43]. They were more satisfied with their lives and reported less stress and financial worries than conventional farmers (Sullivan et al. 1996) [44]. Some positive feelings were associated with conversion, such as pride, freedom, pleasure [46]. Conversely, conventional farming was a source of negative attitudes for the same farmers due to livestock methods and the use of pesticides [46]. Furthermore, organic farmers described their farming as more satisfying, challenging, and better for new experiences and attachment to the farm than conventional farming [42]. Altenbuchner et al. (2017) reported that female organic farmers described financial benefits and improvement in the standard of living by becoming organic farmers, but the persistence of a gender gap concerning income and community [41]. Additionally, more well-being and Buddhist ecological values were reported among organic farmers than conventional farmers by Kaufman and Mock (2014) [45]. Deaconu et al. (2020) also estimated more values among agroecological farmers concerning nutrition and food [39].

**Table 3.** Qualitative studies about organic vs. conventional farmers' psychological health.

| Authors | Location | Farming Production | Sample | Key Findings |
|---|---|---|---|---|
| Altenbuchner et al. (2017) [41] | India | Cotton production | N = 60 Female: $n = 30$ Male: $n = 30$ | Female organic farmers reported financial benefits, a gender gap (concerning income and community), and an improvement in the standard of living. |
| Bouttes et al. (2018) [42] | France | Dairy production | N = 20 | Organic farmers described their farming as satisfying and better for new experiences, challenging by involving new skills, and better for the attachment to the farm. |
| Bouttes et al. (2020) [43] | France-Aveyron | Dairy production | N = 19 | Farmers' satisfaction increased during conversion to organic farming (no effect of conversion strategy). |
| Deaconu et al. (2020, in press) [39] | Ecuador | Fruits, crop, and vegetable production | N = 90 Agroecology: $n = 60$ Conventional: $n = 30$ | Agroecological farmers shared more values concerning food and nutrition. |
| Kaufman and Mock (2014) [45] | Thailand | Rice production | N = 50 Conventional: $n = 5$ Organic: $n = 45$ | Well-being and Buddhist ecological values were higher for organic farmers. |
| Sullivan et al. (1996) [44] | USA-Michigan | Not specified | N = 25 Conventional: $n = 13$ Organic: $n = 12$ | Organic farmers were more satisfied with their lives, less stressed, and less worried by financial considerations than conventional farmers. |
| Van Dam et al. (2010) [46] | France | Not specified | N = 24 | Conversion to organic farming was seen as a way to rationalize negative feelings (e.g., fear, shame, anxiety) due to conventional practices (i.e., use of pesticides and livestock method). Farmers reported positive attitudes (e.g., pleasure, pride, freedom) after their conversion. |



Once again, studies reported differences between farming modes, variables, such as satisfaction, and values were higher on the organic side.

### 3.2.2. Physical Health

Five studies presented qualitative physical health data. Using a similar method to before for psychological health, Altenbuchner et al. (2017) focused on females' perception of organic farming in comparison with conventional farming [41]. Deaconu et al. (2020) collected data about farmers' nutrition [39]. Information about income and health was presented by Beban (2009) and Kaufman and Mock (2014) [45,47]. Finally, Loake (2001) studied one farmer of each type to collect some physical parameters (e.g., heart rate, expended energy) [48].

A total of 259 farmers participated in these studies. There were 46 organic farmers, 60 agroecological farmers, 36 conventional farmers, and 117 who were former conventional farmers who converted to organic. Participants came from diverse countries (e.g., India, Ecuador, Thailand).

Physical health differences were identified by these studies (see Table 4). Indeed, Kaufman and Mock (2014) reported that organic farmers and their families had a better health status than conventional farmers and their families [45]. Beban (2009) also reported an improvement in health and income after converting to organic farming [47]. Moreover, agroecological farmers had a better nutriment adequation than reference farmers due to their production diversity [39]. They also had a better dietary moderation due to the consumption of non-branded food. In Altenbuchner et al.'s (2017) study, an improvement in health condition and food security were reported by female organic farmers in comparison to male organic farmers and to conventional farming [41]. They also reported a higher workload and less training and capacity. The fourth study done by Loake (2001) revealed that the organic participants had a better physical work capacity, a higher physical activity level, and expended more energy than the conventional participant [48]. Moreover, organic farmers' heart rate was more fluctuating than conventional farmers' rate. No difference was found by Loake (2001) for the rate of perceived exhaustion [48].

**Table 4.** Qualitative studies about organic vs. conventional farmers' physical health.

| Authors | Location | Farming Production | Sample | Key Findings |
|---|---|---|---|---|
| Altenbuchner et al. (2017) [41] | India | Cotton production | N = 60<br>Female: *n* = 30<br>Male: *n* = 30 | Female organic farmers reported improvement in health condition, food security but higher workload and less training and capacity. |
| Beban (2009) [47] | Cambodia | Rice and vegetable production | N = 57 | Organic farmers reported an improvement in their health and their income after conversion to organic. |
| Deaconu et al. (2020, in press) [39] | Ecuador | Fruits, crop, and vegetable production | N = 90<br>Agroecology: *n* = 60<br>Conventional: *n* = 30 | Agroecological farmers had better nutriment adequation (due to production diversity), dietary moderation (due to non-marked food consumption). |
| Kaufman and Mock (2014) [45] | Thailand | Rice production | N = 50<br>Conventional: *n* = 5<br>Organic: *n* = 45 | Organic farmers and their families had a better health status. |
| Loake (2001) [48] | UK | All type of farms | N = 2<br>Conventional: *n* = 1<br>Organic: *n* = 1 | The organic farmer had a better physical work capacity, expended more energy, and had a higher physical activity level than the conventional farmer. Their heart rate was more fluctuating than conventional farmers' rate. The rate of perceived exhaustion was similar between farmers. |

Physical health differences were found between organic and conventional farmers, showing a better health condition, nutrition, and physical capacities and activity for organic farmers.

### 3.3. Correlates of Organic Farmers' Health

In the first part, four studies presented quantitative data about correlates of organic farmers' health. Bravo et al. (2012) tested different factors (e.g., farm income, farm management, experience) influence on farmers' satisfaction [49]. The self-reported health of

two categories of organic farmers (i.e., Buddhist temple-based vs. non-profit agriculture) was tested by Kaufman and Mock (2014) [45]. Perrin et al. (2020) looked for predictors of evolutions of farmers' satisfaction and disturbances faced during conversion to organic farming [50]. Rickson et al. (1999) tested factors such as experience with organic practices or feelings about future on satisfaction [32].

A total of 376 organic farmers participated in these studies. They came from Chile, Thailand, France, and Australia.

Twelve correlates of organic farmers' health were identified (see Table 5). Farmers' satisfaction was increased by farm income, positive experience with organic practices, having a 100% organic farm, being optimistic, or feeling good about the future [32,49]. The same studies showed that bureaucratic costs, being depressed, or having little or no debt decreased satisfaction. Reliability farm management, market access, economic costs, and experience were not significant factors [49]. Moreover, the evolution of satisfaction was positively predicted by the initial percentage of maize (before conversion to organic farming) in the utilized agricultural area (UAA) and the evolution of the duration of full grazing [50]. It was yet negatively predicted by the evolution of the percentage of maize (during conversion) in UAA and the initial overall satisfaction [50]. Moreover, disturbances faced during conversion were climatic, economic, health-related, organizational, or technical [50]. Buddhist temple-based farmers reported more health improvement than non-temple-based organic farmers [45].

Farmers' satisfaction and health were affected by some factors such as farm income, the evolution of the duration of full grazing, high positive experience with organic practices, and religion. Among disturbances faced during conversion to organic farming, some of them were health-related. Twelve Cohen's *d* were calculated from these results. They varied from 0.39 to 0.98. Four effects were small, six were moderate, and two were large.

In the second part, three studies reported qualitative health data about organic farmers. Important values for well-being were inventoried by Beban (2009) [47]. Brigance et al. (2018) focused on perceived risks and protective factors of organic farming [51]. Soto Mas et al. (2017) also studied risks linked to small farms [52].

A total of 117 organic farmers participated in these studies. They came from Cambodia and the USA.

As depicted in Table 6, farmers' well-being was related to growing their own resources for their family, being healthy, having enough money and enough to eat, and a clean environment [47]. Perceived risks were loss, work-life balance, workload, isolation, and dependence on their environment [51]. For small farms, stress, social difficulties, and isolation were associated risks, especially [52]. Protective factors of organic farming were contentment, satisfaction, pride, social networks and interconnections, cohesion, sense of community, and responsibility [51].

**Table 5.** Quantitative studies about correlates of organic farmers' health.

| Authors | Location | Farming Production | Sample | Key Findings | Cohen's *d* |
|---|---|---|---|---|---|
| Bravo et al. (2012) [49] | Chile | Fruits and vegetable farming, crop production | N = 60 | Organic farmers' satisfaction was increased by farm income ($p < 0.01$) and decreased by bureaucratic costs ($p < 0.05$). Reliability, farm management, market access, economic costs, and experience did not affect satisfaction ($p > 0.10$). | 0.45 (Farm income on satisfaction) 0.43 (Bureaucratic costs on satisfaction) |
| Kaufman and Mock (2014) [45] | Thailand | Rice production | N = 75 Temple-based: *n* = 36 Non-profit: *n* = 39 | Temple-based farmers self-reported more health improvement ($p = 0.003$). | 0.72 (Farming method on self-reported health) |
| Perrin et al. (2020) [50] | France | Dairy production | N = 81 | The initial overall satisfaction and the evolution of the percentage of maize in the utilized agricultural area (UAA) negatively predicted the evolution of farmers' satisfaction during conversion. The initial percentage of maize in UAA and the evolution of the duration of full grazing were positively linked with satisfaction. Disturbances faced in the farms were climatic, economic, health-related, organizational, or technical. | 0.90 (Initial overall satisfaction on the evolution of satisfaction) 0.98 (Evolution of percentage of maize in UAA on the evolution of satisfaction) 0.70 (Initial percentage of maize in UAA on the evolution of satisfaction) 0.54 (Evolution of the duration of full grazing on the evolution of satisfaction) |
| Rickson et al. (1999) [32] | Australia | Crop and livestock production | N = 282 Conventional: *n* = 122 Organic: *n* = 160 | High positive experience with organic practices ($p = 0.003$), being bright or good about future ($p < 0.001$) and having a 100% organic farm ($p = 0.01$) increased farmers' work satisfaction. Being depressed about future ($p = 0.02$) and having little or no debt ($p = 0.05$) decreased work satisfaction. | 0.56 (High positive experience on satisfaction) 0.68 (Being bright or good about future on satisfaction) 0.47 (Having a 100% organic farm on satisfaction) 0.50 (Being depressed about future on satisfaction) 0.39 (Having little or no debt on satisfaction) |

Cohen's *d*: 0.45 and 0.43 calculated with Procedure 5; 0.72 calculated with Procedure 6; 0.56, 0.68, 0.47, 0.50, and 0.39 calculated with Procedure 1; 0.90, 0.98, 0.70, and 0.54 calculated with Procedure 7.

**Table 6.** Qualitative studies about correlates of organic farmers' health.

| Authors | Location | Farming Production | Sample | Key Findings |
|---|---|---|---|---|
| Beban (2009) [47] | Cambodia | Rice and vegetable production | N = 57 | Reported important values for organic farmers' well-being were to grow his own resources for his family, have good health, enough money, and enough to eat, and a clean environment. |
| Brigance et al. (2018) [51] | USA–New Mexico | Crop production | N = 30 Producer: *n* = 10 Worker: *n* = 20 | Loss, work-life balance, workload, isolation, and dependence on the environment were perceived as risks (similar to conventional risks) by farmers. Contentment, satisfaction, pride, interconnections, social networks, cohesion, sense of community, and responsibility were seen as protective factors. |
| Soto Mas et al. (2017) [52] | USA–New Mexico | USDA-certified produced/small farms | N = 30 Producer: *n* = 10 Worker: *n* = 20 | Small farms were associated with risk, stress, social difficulties, and social isolation. |

Thirteen positive correlates to organic farmers' health were identified (e.g., having good health, satisfaction, social networks), and nine negative factors (e.g., loss, workload, social difficulties). Nine of these correlations were linked to the social domain.

In the third part, two studies presented data about correlates of conventional farmers' health. Similar to organic farmers, Rickson et al. (1999) measured predictors of farmers' work satisfaction [32]. Siegel et al. (2017) presented data about the effects of pesticide exposure on depression, measured by the CES-D [53].

A total of 814 conventional farmers participated in these studies. They came from Australia and the USA.

Both studies reported effects (cf. Table 7). Feelings about the future affected work satisfaction: being depressed decreased it but being optimistic or feeling good increased satisfaction [32]. In addition, exposure to any solvent, gasoline, or petroleum distillates increased depression scores on the CES-D, but exposure to paint thinner did not have an effect [53].

**Table 7.** Quantitative studies about correlates of conventional farmers' health.

| Authors | Location | Farming Production | Sample | Key Findings | Cohen's *d* |
|---|---|---|---|---|---|
| Rickson et al. (1999) [32] | Australia | Crop and livestock production | N = 282 Conventional: *n* = 122 Organic: *n* = 160 | Being depressed about the future decreased conventional farmers' work satisfaction ($p = 0.05$). Being good or bright about the future increased work satisfaction ($p < 0.001$). | 0.39 (Being depressed about future on satisfaction) 0.93 (Being bright or good about future on satisfaction) |
| Siegel et al. (2017) [53] | USA-Iowa and North Carolina | Different types of systems | N = 692 | Exposition to any solvent ($p = 0.04$), gasoline ($p = 0.04$), or petroleum distillates ($p = 0.03$) was linked to a higher depression score on the CES-D (Center for Epidemiologic Studies Depression Scale). Paint thinner did not have significative effect ($p > 0.10$). | 0.24 (Exposition to any solvent on depression) 0.24 (Exposition to gasoline on depression) 0.26 (Exposition to petroleum distillates on depression) |

Cohen's *d*: 0.39, 0.93, 0.24, 0.24, and 0.26 calculated with Procedure 1.

Five correlates to conventional farmers' health were identified, two linked with emotions and three linked to pesticides. Different Cohen's *d* were calculated for each effect; they varied from 0.24 to 0.93. Four effects were small, and one was large.

## 4. Discussion

The aim of this article was to review literature comparing organic, agroecological, and conventional farmers' health and its correlates. Results from 29 relevant articles indicated numerous differences between farmers. Concerning psychological health, most of these studies showed that organic farmers were in a better condition when looking at measures

of depression, anxiety, satisfaction, well-being, etc. [13,17,28,29,31,32,39,42–46]. Other studies did not find differences between organic and conventional farmers for psychological indicators [17,28,30]. Similarly, physical health data usually presented better results among organic farmers than conventional farmers [18,34–37,39,40,45]. Rarely, no differences or advantages for conventional farmers were reported [17,28,29,39]. Thus, a large majority of results presented better psychological and physical health among organic farmers; little or no differences were identified in favor of conventional farmers. Very few studies dealt with agroecological farms and are considered organic. Concerning correlates of farmers' health, psychological, social, financial, and agricultural factors were often highlighted [31,45,47,49–53]. Financial factors question the higher pay among organic farmers and their better health. Near half of Cohen's *d* calculated from these results indicated small effects, a third indicated intermediate effects, and a quarter indicated large effects. These parameters have to be interpreted carefully because of the heterogeneity of dependent variables used in all studies.

### 4.1. Work Satisfaction as an Indicator of Health and Well-Being

An original result with an important social dimension is that organic farmers have higher job satisfaction than conventional farmers (Mzoughi, 2014). This aspect reminds us that good health is defined as much by the absence of disease as by a state of mental and psychological well-being [54]. Our article invites researchers to learn more about the role of well-being in the professional practice of agriculture. Work carried out in the field in different disciplines of the human and social sciences describes and analyzes the positive dynamics for the organic farmer generated by well-being: a more peaceful relationship with nature [55], the reconsideration of the links of domination between humans and animals [56], the reformulation by the farmers themselves of the values and meaning they give to their profession [46]. In order to deepen our knowledge of the psychological mechanisms highlighted in the literature, it would be appropriate to test the question of job satisfaction as a predictor of well-being and health.

### 4.2. A Social Dimension to Be Coupled with an Agronomic Dimension

The qualitative results question the social dimensions of agricultural activity. Thus, a study conducted among women farmers in India showed that the practice of agroecology allows for better financial remuneration and improves the standard of living of women. Other studies pointed out that organic farming requires specific technical skills which, in turn, stimulate farmers and strengthen attachment to their farm. This dynamic is particularly present when converting to organic farming. However, these results focused on a specific category of farmers (the organic). It is, therefore, necessary to consider the agronomic dynamics involving organic and conventional farmers. This aspect is addressed in the analysis of the literature that we carried out, but more qualitative research could be implemented because it would provide elements of knowledge in agronomy and sociology. Thus, the typology of farmer profiles developed by Darnhofer et al. (2005) highlights the agronomic and economic strategies of farmers in a situation of change or not towards organic agriculture [16]. The author distinguished five profiles: the "committed conventional", the "pragmatic conventional", the "environment-conscious but not organic", the "pragmatic organic", the "committed organic". A complementary study based on interviews and observations would shed light on the psychological motivations for choosing or not choosing one or the other farming systems [15,46].

### 4.3. Associate with a Societal Dimension

The societal dimension of health issues in agriculture should be discussed too. Some examples taken from the quoted articles illustrate it. A qualitative study conducted by Deaconu et al. (2020) showed how agroecological farmers in Ecuador more widely share common values of food and nutrition common to the whole society [39]. Kaufman and Mock's work (2014) highlighted results linked to concerns raised by organic farmers, and which make sense for the whole of Thai society—the religious approaches to the ecological

question, the improvement in social status following the practice of organic, the risks of isolation of small farmers [45]. The analysis of psychological mechanisms of conversion as formulated by Van Dam et al. (2010) demonstrated how the choice to go organic is based on the transition from emotions of shame and fear—linked to the use of dangerous chemicals—to positive emotions of pride and pleasure [46]. More generally, it is worth recalling the structural link in contemporary societies between the emergence of organic agriculture and the criticism of productivism and industrial agriculture. Likewise, the issue of farmers' health comes up in societal debates on the environment, sustainable development, and climate change.

## 5. Conclusions

This review of the literature showed that organic farming has a better overall effect on farmers' self-reported mental health and biological markers of physical health. Moreover, factors correlated with farmers' health were usually psychological, social, financial, and agricultural. This analysis also highlighted the small number of studies on this subject and encourages further research in this area for different agricultural systems, including the agroecological model, which is developing strongly throughout the world.

**Author Contributions:** Conceptualization, L.D., M.D., M.S. and A.M.; Methodology, L.D., M.D, M.S. and A.M.; Software, L.D., M.D, M.S. and A.M.; Validation, L.D., M.D., M.S., R.H. and A.M.; Formal Analysis, L.D., M.D., M.S., R.H. and A.M.; Investigation, L.D., M.D., M.S., R.H. and A.M.; Resources, L.D., M.D., M.S., R.H. and A.M.; Data Curation, L.D., M.D., M.S., R.H. and A.M.; Writing–Original Draft Preparation, L.D., M.D., M.S., R.H. and A.M.; Writing–Review & Editing, L.D., M.D., M.S., R.H. and A.M.; Visualization, L.D., M.D., M.S., R.H. and A.M.; Supervision, M.D., M.S. and A.M.; Project Administration, M.D., M.S. and A.M.; Funding Acquisition, M.D., M.S. and A.M. All authors have read and agreed to the published version of the manuscript.

**Funding:** This research received no external funding.

**Institutional Review Board Statement:** Not applicable.

**Informed Consent Statement:** Not applicable.

**Data Availability Statement:** All the data comes from the literature and is accessible with the key words mentioned in materials and methods.

**Conflicts of Interest:** The authors declare no conflict of interest.

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
