# Peer review of "Psychological and Physical Health of Organic and Conventional Farmers: A Review"

_sustainability, doi:10.3390/su132011384_

Round 1
Reviewer 1 Report
Dear Authors,
I found your paper interesting and I think it has certain potential to contribute to Sustainability but there are quite some issues to reconsider. Below I have some comments for the Authors to address:
Title
I am not truly convinced that the title “Psychological and Physical Health of Organic and Conventional Farmers: A Review” is the most appropriate. The review covers mainly with studies with self reported psychological and physical health. I would suggest reconsidering the title not to make false premises.
Abstract
Effect sizes were small to large; they differed according to the target outcome. Also, factors correlated with farmers’ health were usually psychological, social, financial, and agricultural. Conclusions are discussed regarding the conception of health, types of farming production and intergroup relations.
Introduction
What I miss in the introduction is a comprehensive overview of the impact of socio-economic factors on farmers health. Authors should reflect on it that the organic farmers are often better educated and more inclined to engage in certain behaviours that potentially result in better psychological and physical health. Another issue are related to the social environment. Organic farmers are often better organized and can experience more social support that in turn may positively affect their psychological health. Authors should go beyond focusing in the introduction entirely on differences related to farming methods or as stated “agricultural model”.
Method
L 64 “In this review, included articles had to focus on the comparison between organic (grouping agroecological, ecological and organic farming) and conventional farmers’ psychological or physical health”. Clumsily written and needs revision. Please specify what you mean by agroecological farming? What is the difference between ecological and organic in your review? According to EU regulation on organic farming these are synonyms.
The PRIMSA diagram showing the literature search and selection process should be presented in research methods. The Authors should describe in a comprehensive manner how they adhered to PRISMA guidelines.
Statistical procedures
I would recommend to describe more in detail the approach to statistical analyses before placing the information on Cohen’s d.
Discussion
There is no clear line of discussion. I would suggest to structure the discussion dividing it into subchapters.
The second part of the discussion needs particular attention because some paragraphs may lead to misconception e.g.:
L346 “This review allowed us to focus on the differentiation processes of the agronomic system between organic and conventional farmers” It is unclear why you make a reference to the differentiation of the agronomic system as an outcome of the review.
What I miss in the review is any concluding remarks.
Author Response
Dear reviewer
we have taken into account all the remarks made on the document. You will find enclosed the answers to the remarks of the three reviewers.
Regards
Audrey Michaud

Reviewer 2 Report
Dear Authors, thank you for the opportunity to review this interesting review manuscript. The paper is filling a lack of knowledge overview in the current scientific literature about the psychological and physical health of organic and conventional farmers.
The aim and the objectives are clear. In general, I can say that the review paper brings very useful information of previously published research on the topic, especially for policy makers.
Although the authors rely on the predigital era practice of reading papers for surfacing their conceptual and theoretical contributions, the relevant scientific literature is very well reviewed and discussed.
The paper demonstrates an adequate understanding of the proper knowledge about the possible effect of the agricultural model (organic or conventional farming) on farmer’s psychological and physical health, in correct connection with results and suggestions. However , some statements in the paper should be supported by more evidence and relevant references (eg “Organic farming is often mentioned as a solution to these health problems”, “Farmers’ satisfaction and health were affected by some factors such as farm income, evolution of the duration of full grazing, high positive experience with organic practices, and religion...”, «Thirteen positive correlates to organic farmers’ health were identified ..., and nine negative factors ...», «Nine of these correlations were linked to the social domain», «The qualitative results question the social dimensions of agricultural activity»).
The findings are remarkably interesting.
Results are clearly presented, and the discussion provide a better understanding of them, which are consistent with the development and the findings of the study. The practical implications of the research show an in-depth analysis of the research literature and the objective. However, little has been written on the part of extending beyond the literature search and summary of past research to the development of theoretical directions for the future.
The paper has good quality of communication, in general is well structured (the tables are excepted) and quite easy to read.
I agree with the Authors that the agricultural sector is sensitive in terms of the farmers’ health and the effects of agricultural models on the psychological and physical farmers’ health remains not clear and requires a synthesis of the literature.
There are many gaps in the current research on this specific field, and the review paper is worth publishing after following minor revisions to the conclusion (theoretical directions for the future), the tables style and references.
Author Response
dear reviewer,
we have reworked the whole document taking into account the different remarks. You will find attached a detailed response to each comment.
Sincerely
Audrey Michaud

Reviewer 3 Report
The authors review the literature that compares conventional and organic farmers in terms of their psychological and physical health.
The article is relevant and timely. The paper is straightforward, well-written and well structured. Methods are appropriate and adequately described. Tables are informative and clearly presented. The discussion and conclusions are supported by the results.
I have some comments which I list below:
Broad comments:
- The paper´s contribution to the literature needs to be better worked out/refined.
- Regarding the selection method, I think it is appropriate and is adequately described as stated before. If anything, I would suggest including a summary table on the article’s selection process (up to the authors).
- Regarding the statistical procedure, I would suggest explaining what the Cohen’s d measures and why it is used here. I would also suggest including a short note to explain why different procedures are necessary and if using different procedures may have any implications. I would suggest explaining the Cohen’s d measure´s interpretation here too.
Specific comments:
- Line 43: I would suggest inserting a reference/references for the following statement: “Organic farming is often mentioned as a solution to these health problems”.
- Tables 1-7: I would suggest including the reference number after the names’ authors. For example, the two references Nankongn et al (2020) can lead to confusion.
- Line 91: Only for qualitative data?
- Lines 140-141: I think Table 1 does not show the following: “mental health wasn’t affected by type of farming”. The table states that “Organic farmers were less depressed than conventional farmers on the SDHS”.
- Table 1 (Howard et al): 77 and 42 does not add up 125.
- Line 161: The authors state that “there were 2892 organic farmers, 60 agro-ecological farmers, 37 ecological farmers”. I think the numbers do not match with those shown in Table 2. Please check it and correct it if necessary.
- Line 162: I think the number in “3522 conventional farmers” does not match the total number in Table 2. Please check it and correct it if necessary.
- Line 165: Costa et al. (2021) is Costa et al. (2014) according to Table 2. Please correct it where necessary.
- Line 188: I think they are 11 (according to Table 2). Please correct it where necessary.
- Table 2 (Deaconu et al): The table says that the year of publication is 2021 but the references say 2020. The same applies to Table 3.
- Table 3: Does “reference” refers to conventional? If so, I would suggest changing it for conventional. Otherwise, it leads to confusion. The same applies to Table 4.
- Line 227: I think it is [38]. Please check it and correct it if necessary.
- Line 256: Which two categories? I would suggest specifying them.
- Line 261: I think the number 376 does not match the total Ns in Table 5. I would suggest checking this and correcting if necessary.
- Lines 268-271: How can “the evolution of satisfaction was positively predicted by percentage of maize in utilized agricultural area (UAA) during conversion … but it was negatively predicted by the percentage of maize in UAA” at the same time? I would suggest clarifying this.
- Line 282: Is the reference number [31] included at the end of this sentence correctly written here?
- Line 304: I think the number 814 does not match the total Ns in Table 5. I would suggest checking this and correcting if necessary.
- Line 314: Is the sentence correctly referenced? I think it does not need reference.
Author Response

(The authors gave the same response as above.)

Round 2
Reviewer 1 Report
Dear Authors,
I think the paper has been sufficiently improved. My only concern is the conclusion section “This review of the literature showed that organic farming has a better overall effect on farmers' health”. Definitely, the Authors should underline in the conclusion section that it refers to self reported data.
Author Response
Dear reviewer,
we have taken into account your remark and specified this element in the concluding sentence: This review of the literature showed that organic farming has a better overall effect on farmers' self-reported mental health
Regards
Audrey Michaud